# Impact of SARS-CoV-2 Infection on Unvaccinated Pregnant Women: Non-Reassuring Fetal Heart Rate Tracing Because of Placentitis

**DOI:** 10.3390/v15051069

**Published:** 2023-04-27

**Authors:** Alexandra Claudet, Daniele De Luca, Elie Mosnino, Jérémie Mattern, Olivier Picone, Jeanne Sibiude, Estelle Wafo, Vassilis Tsatsaris, Emilie Giral, Irène Grefenstette, Julie Carrara, Dominique A. Badr, Marie-Hélène Saint-Frison, Sophie Prevot, Alexandra Benachi, Alexandre J. Vivanti

**Affiliations:** 1Division of Obstetrics and Gynecology, “Antoine Béclère” Hospital, Paris Saclay University Hospitals, APHP, 75000 Paris, France; 2Division of Pediatrics and Neonatal Critical Care, “Antoine Béclère” Hospital, Paris Saclay University Hospitals, APHP, 75000 Paris, France; 3Division of Obstetrics and Gynecology, “Louis Mourier” Hospital, Paris Nord Val de Seine University, APHP, 75000 Paris, France; 4Inserm IAME 1137, 75000 Paris, France; 5GRIG: Groupe de Recherche sur les Infections en cours de Grossesse, 75000 Paris, France; 6Division of Obstetrics and Gynecology, “Marne La Vallée” Hospital, 77600 Jossigny, France; 7Division of Obstetrics and Gynecology, “Port Royal” Hospital, Paris Centre University Hospitals, APHP, 75000 Paris, France; 8Division of Obstetrics and Gynecology, “André Grégoire” Hospital, 93100 Montreuil, France; 9Division of Obstetrics and Gynecology, “Centre Hospitalier des Quatre Villes”, 92064 Saint Cloud, France; 10Division of Obstetrics and Gynecology, “University Hospital Brugmann”, Université Libre de Bruxelles, 1070 Brussels, Belgium; 11Division of Foetopathology, “Robert Debré” Hospital, APHP, 75000 Paris, France; 12Division of Pathology, “Bicetre” Hospital, Paris Saclay University Hospitals, APHP, 94270 Le Kremlin-Bicêtre, France

**Keywords:** COVID-19, infection, coagulopathy, pregnancy, fetus, acidosis, placentitis, transmission

## Abstract

In 2020, a new coronavirus, called severe acute respiratory syndrome coronavirus 2 (SARS-CoV-2), emerged in China. SARS-CoV-2 infection has been shown to be highly morbid in pregnant women, being a risk factor for several obstetric conditions leading to increased maternal and neonatal mortality. A few studies since 2020 have shown SARS-CoV-2 maternal–fetal transmission and noted placental abnormalities grouped under the term placentitis. We hypothesized that these placental lesions could be responsible for abnormalities in placental exchange and therefore abnormalities in cardiotocographic monitoring, leading to premature fetal extraction. The objective is to identify the clinical, biochemical, and histological determinants associated with the occurrence of non-reassuring fetal heart rate (NRFHR) outside labor in fetuses of SARS-CoV-2-infected mothers. We conducted a retrospective multicenter case series of the natural history of maternal SARS-CoV-2 infections resulting in fetal delivery outside labor due to NRFHR. Collaboration was sought with the maternity hospitals in the CEGORIF, the APHP and Brussels hospitals. The investigators were contacted by e-mail on three successive occasions over a period of one year. Data from 17 mothers and 17 fetuses were analyzed. Most women had a mild SARS-CoV-2 infection; only two women presented severe infection. No woman was vaccinated. We found a substantial proportion of maternal coagulopathy at birth: elevation of APTT ratio (62%), thrombocytopenia (41%) and liver cytolysis (58.3%). Iatrogenic prematurity was noted in 15 of 17 fetuses, and 100% were born by cesarean delivery due to emergency criteria. One male neonate died on the day of birth due to peripartum asphyxia. Three cases of maternal–fetal transmission were recorded following WHO criteria. Placental analysis in 15 cases revealed eight cases of SARS-CoV-2 placentitis, causing placental insufficiency. In total, 100% of the placentas analyzed showed at least one lesion suggestive of placentitis. SARS-CoV-2 maternal infection during pregnancy is likely to generate neonatal morbidity in relation to placental damage resulting in placental insufficiency. This morbidity may be the consequence of induced prematurity as well as acidosis in the most severe situations. Placental damage occurred in unvaccinated women and in women with no identified risk factor, in contrast to severe maternal clinical forms.

## 1. Introduction

Severe acute respiratory syndrome coronavirus 2 (SARS-CoV-2) infection has been shown to be highly morbid in pregnant women, being a risk factor for several obstetric conditions (pre-eclampsia, prematurity, cesarean delivery) and leading to increased maternal and neonatal mortality in the case of severe infection [1,2,3]. Similarly, pregnancy after 20 weeks is an independent risk factor for developing a severe form of SARS-CoV-2 infection [4].

A systematic review showed that most newborns infected with SARS-CoV-2 were pauci-symptomatic, although rare severe cases may occur. Several cases of vertical transplacental transmission have been reported and have demonstrated that maternal–fetal infections are possible, although rare. Vertical transplacental transmission of SARS-CoV-2 is defined by the World Health Organization (WHO) by several criteria: maternal infection during pregnancy, in utero exposure to SARS-CoV-2, and persistence of infection or an immune response against SARS-CoV-2 in the newborn [5]. Maternal–fetal transmission by the transplacental route was demonstrated by Vivanti et al. in 2020 in a 23-year-old infected woman who underwent cesarean section at 35^5/7^ for fetal heart rhythm abnormalities [6]. SARS-CoV-2 testing was positive in amniotic fluid, placenta, fetal serology and fetal bronchoalveolar fluid. The placenta showed diffuse chronic intervillositis, diffuse perivillous fibrinous deposits, and ischemic nodules. In another study by Vivanti et al. reporting six cases of transplacental transmission of SARS-CoV-2, fetal distress at birth was related to the extent of placental inflammation, associated with neonatal acidosis and an increased rate of admission to neonatal intensive care [7]. A study published in 2021 reported chronic intervillositis lesions and fibrin deposits in the placentas of SARS-CoV-2-infected neonates [8]. The most severe perinatal impact does not seem to be related to congenital infections but rather to the significant increase in stillbirths, especially linked to the Delta strain [9]. A large study has also shown an association between stillbirths related to SARS-CoV-2 infection, chronic intervillositis and massive fibrin deposits [10]. Finally, a few case reports and numerous anecdotal data have reported non-reassuring fetal heart rate (NRFHR) tracing outside labor in SARS-CoV-2-infected women [6,11,12]. NRFHR may be responsible for preterm fetal delivery. Although no causal link has been made to date, we hypothesized that the placental lesions observed could be responsible for abnormalities in placental exchange and therefore abnormalities in the cardiotocographic (CTG) monitoring leading to premature fetal extraction and to stillbirth in the most severe cases.

The aim of our study was to identify the clinical, biochemical, and histological determinants associated with the occurrence of NRHFR outside labor in fetuses of SARS-CoV-2-infected mothers.

## 2. Methods

### 2.1. Study Design

This was a retrospective observational multicenter study (case series) reporting the natural history of maternal SARS-CoV-2 infections resulting in fetal delivery outside labor due to NRFHR. Collaboration was sought with the maternity hospitals in the CEGORIF (Cercle d’Étude des Gynécologues-Obstétriciens de la Région Ile-de-France), the APHP (Assistance Publique des Hôpitaux de Paris) and Burgmann University (Brussels) hospitals. The investigators were contacted by e-mail on three successive occasions over a period of one year.

### 2.2. Eligibility and Exclusion Criteria

Women were included between December 2020 and December 2021.

Inclusion criteria were:−age ≥ 18 years old;−maternal SARS-CoV-2 infection confirmed by RT-PCR on nasopharyngeal swab according to routine care;−delivery because of NRHFR tracing, after 24 weeks of pregnancy;

Exclusion criteria were:−in utero fetal death;−placenta abruptio or ongoing severe preeclampsia [13];−Benckiser hemorrhage;−uterine rupture;−painful uterine contractions or labor in progress.

### 2.3. Data

Anonymized data were retrospectively extracted from patients’ medical records and transmitted to the principal investigator via a case report form. Only the investigators involved in the data analysis had access to the complete data collection form previously anonymized. All the data collected were routine care data that did not require any additional consultation. Neonatal birthweights were classified at birth according to the AUDIPOG curves [14].

Placentas were analyzed by every pathologist from the different hospitals according to the Amsterdam Consensus statement [15]. They conducted a classical pathology examination of each placenta they received, and then prepared and stained tissues for immunohistochemistry/immunofluorescence investigation according to routine care. Some of them completed their analyses with anti-SARS-CoV-2 antibodies against placental tissues.

### 2.4. Statistical Analyses

The database was completed in an Excel workbook (Microsoft Corporation, Redmond, WA, USA) and anonymized with an alphanumeric code. Cumulative estimates of event rates (frequency) were reported as a percentage (%), which refers to the total number of fetuses/neonates, unless otherwise indicated. Continuous data were described as median (interquartile range). All statistical analyses were performed using R software (R Core Team, version 3.5.2).

### 2.5. Ethics Statement

The study was approved by the appropriate ethics board on the 8 June 2021 (CEROG 2021-OBST-0503). Informed consent was waived because of the retrospective nature of the study and the analysis used anonymous clinical data.

## 3. Results

During the study period, 17 women from seven different maternity hospitals were included (CHU Cochin Port-Royal in Paris, CH Marne La Vallée in Josigny, CHU Antoine Béclère in Clamart, CHU Louis Mourier in Colombes, CH Montreuil, CH de Saint-Cloud, CHU Brugmann). The clinical characteristics of the women are listed in Table 1. All women had singleton pregnancies. Four (23%) women had a BMI over 30 kg/cm^2^ before pregnancy. Nine women (53%) were of Caucasian origin, and six (35%) were of African origin. Most of the women did not have any comorbidity, but it should be noted that two (12%) women had liver disease before pregnancy (hepatic steatosis and chronic hepatitis B). In addition, three (18%) women had controlled asthma without recent hospitalization. No woman was vaccinated against SARS-CoV-2. Pregnancies had few obstetric pathologies: two (12%) pregnancies were complicated by gestational diabetes, and one (6%) pregnancy was complicated by moderate preeclampsia and gravida cholestasis. All fetuses were considered eutrophic at ultrasound follow-up, without morphological anomalies (all women had regular three-monthly ultrasound follow-up according to the recommendations of French scientific societies).

The results of clinical characteristics of SARS-CoV2 infection are presented in Table 2. Women were tested for SARS-CoV-2 around 33 weeks (median 32^6/7^ weeks, IQR (30^5/7^–35^2/7^), min 26, max 37^2/7^) because of symptoms. According to the NIH classification (Appendix A) [16], 12 (71%) women had mild SARS-CoV-2 infection, three (18%) had moderate infection, and two (12%) had severe infection. None of them had a critical infection. The main symptoms experienced by the women were fever (*n* = 12; 71%), cough (*n* = 8; 47%), body aches and dyspnea (*n* = 3; 18%), ageusia and anosmia (*n* = 2; 12%), headache (*n* = 1; 6%) and digestive disorders (*n* = 1; 6%). A viral strain identification was performed and reported for seven women: wild type (Wuhan-China; one case), B.1.1.7 “Alpha” (UK strain; five cases) and B.1.617.2 “Delta” (India; one case).

Only two (12%) women of the cohort consulted for an obstetrical concern (decreased fetal movements) but were also symptomatic. All patients were hospitalized with a median time from the day of diagnosis to the day of hospitalization of four days (IQR 2–6), at a median term of 33^5/7^ weeks (IQR (31^3/7^–35^6/7^)). Two (12%) patients were admitted to the ICU for respiratory distress related to severe SARS-CoV-2 infection: one (6%) woman presented with fetal bradycardia in the ICU and was intubated to perform an emergency cesarean delivery, and the second woman was intubated and extubated between 27 and 28 days after birth. It should be noted that for five (29%) women the day of diagnosis was the same as the day of hospitalization: four of them had fever associated with biochemical abnormalities (thrombocytopenia, lymphocytopenia, prolonged APTT and elevated liver enzymes) and the last one presented NRFHR as an obstetrical emergency requiring immediate cesarean delivery. Most women (88%) required simple inpatient monitoring associated with regular blood tests during hospitalization, but two (12%) women were admitted prepartum to the ICU for respiratory distress related to severe SARS-CoV-2 infection.

The data of obstetrical and neonatal characteristics are shown in Table 3. All women delivered by cesarean delivery. None of the NRHFR tracings allowed expectant management or induced vaginal delivery. The women delivered at a median term of 33^6/7^ weeks (IQR (31.2–35.4)). A total of 88% of women gave birth prematurely: one woman before 32 weeks, five women between 28 and 32 weeks and nine women between 32 and 37 weeks. The last two women had a cesarean delivery at 37^3/7^ weeks and 37^4/7^ weeks. Half of the hospitalized women gave birth within five days of hospitalization.

Concerning the neonates, the median birthweight Z-score was −0.13 (IQR (−1.3–0.11)). Five (29%) neonates had a birthweight below the 10th percentile and two (12%) of them below the 5th percentile. For 40% of the newborns, the 5 min Apgar score was lower than 7. Six (40%) cases of moderate (arterial cord pH between 7.0 and 7.2) and one (7%) case of severe (arterial cord pH < 7.0) acidosis were observed. Twelve newborns (71%) were immediately admitted to the neonatal ICU (NICU), 11 of them (65%) because of respiratory distress. One neonatal death at 24 min of life occurred in the NICU (delivery at a gestational age of 27^2/7^ weeks).

Seven (41%) newborns were tested at birth for SARS-CoV-2 (nasopharyngeal swab): three (43%) were positive and four (57%) were negative.

The results of maternal biochemical characteristics are presented in Figure 1 and Appendix A. Thrombocytopenia was observed in seven patients (41%) at birth vs. four (24%) at hospitalization (median 136 G/L; IQR (106–218) at hospitalization and 122 G/L; IQR (62–218) at birth). Lymphocytopenia was also observed in most patients at hospitalization (63%). We also observed a prolonged APTT in patients during hospitalization (69%) that persisted on the day of fetal delivery. Most patients had elevated liver enzymes (>2N) at birth (58%). Regarding fibrinogen levels, the trend was downward from diagnosis (median 4.3 g/L, IQR (2.3–5.1)) to the day of delivery (median 2.8 g/L, IQR (1.2–4.3)).

Placental characteristics are provided in Table 3. Two placentas were missing because they were not sent for analysis after birth. Eight placentas out of fifteen were tested for SARS-CoV-2 and all were positive for SARS-CoV-2 infection (RT-PCR or SARS-CoV-2-specific immunostaining). In addition, we collected placental pathology analysis of 15 placentas: In 67% of placentas, lesions of chronic intervillositis and fibrinoid deposits were observed. A total of 60% of placentas showed trophoblast necrosis. A total of 100% of the placentas analyzed showed at least one of these lesions. Within the eight positive placentas for the SARS-CoV-2, we collected all placental histologic analyses: intervillositis (7/8), trophoblast necrosis (7/8) and fibrinoid deposits (6/8). Concerning the deceased neonate, the placenta showed only intervillositis and no SARS-CoV-2 search was performed.

## 4. Discussion

This retrospective observational multicenter study reports the natural history of maternal SARS-CoV-2 infections resulting in fetal delivery outside labor due to NRFHR. We found, in pregnant women with mild SARS-CoV-2 infection, consistent biochemical abnormalities as thrombopenia and liver cytolysis. In addition, all tested placentas were infected with SARS-CoV-2, in most cases associated with severe inflammatory lesions. These placental injuries are a potential explanation of NRFHR resulting from the fetal hypoxemia and neonatal acidosis observed in our study.

It should be noted that 70% of the women in the cohort had a mild infection and the majority had no identifiable risk factors. The occurrence of inflammatory lesions that result in NRFHR thus appears to be independent of the severity of maternal infection. In addition, almost one in two patients had a coagulation disorder at the time of birth. Coagulopathy induced by SARS-CoV-2 infection was described as early as 2020, associated with HELLP-like syndromes. Indeed, in infected pregnant patients, hemostasis disorders with prolonged APTT, thrombocytopenia and association with hepatic cytolysis have been described [18]. These results were also found in our population, without distinction with the severity of the SARS-CoV-2 infection, in agreement with previous studies [11,18,19]. Some teams have described hepatic cytolysis in patients infected with SARS-CoV-2 suggestive of HELLP-like syndromes [19]. It should also be noted in our study that cytolysis seemed to increase progressively from diagnosis to birth. Moreover, fibrinogen levels tended to decrease during hospitalization, which is consistent with the coagulopathy profile previously described during SARS-CoV-2 infection [18].

Koumoutsea et al. first reported in 2020 the existence of a potential link between SARS-CoV-2 infection during pregnancy and the resulting complications of coagulopathy, NRHFR and placental histological lesions [18]. Sichitiu et al. have identified 11 cases since 2020 of SARS-CoV-2 infections classified as “mild” or “moderate” according to the WHO classification in which there was NRHFR outside labor, with emergency extraction by cesarean delivery [20]. One of the clinical similarities found in these case reports is decreased active fetal movements. This sign has been previously described by Favre et al. [5]. In our case-series, only two women were seen at first for decreased active fetal movement.

SARS-CoV-2 has been previously described as a major risk factor for stillbirth [10,21]. We can hypothesize that the existence of NRFHR in association with inflammatory placental lesions may represent the stage that precedes stillbirth by asphyxia. In the case of maternal infection with SARS-CoV-2, the existence of biochemical abnormalities (transaminitis, coagulation abnormalities) could lead to consideration of the implementation of CTG monitoring to limit the risk of stillbirth.

No woman in our study was vaccinated, even among the most recent women included in December 2021. We reissued a call for observations in August 2022 with the same inclusion and exclusion criteria as initially and no cases were reported. This absence of new cases of SARS-CoV-2 placentitis can be explained by the appearance and democratization of anti-SARS-CoV-2 vaccination in pregnant women from April 2021 in France and the spread of recommendations. According to an Australian retrospective study, vaccinated patients had a significantly lower rate of stillbirth compared with unvaccinated patients, a significant reduction in total preterm births < 37 weeks, spontaneous preterm birth and iatrogenic preterm birth [22]. The impact of vaccination in pregnant women may have led to a decrease in maternal-neonatal transmission of SARS-CoV-2, but other studies should be conducted to confirm this observation. Furthermore, the appearance of the Omicron variant in France from December 2021 seems to be less obstetrically pathogenic and less likely to cause maternal–fetal transmission: maternal morbidity and mortality decrease in the case of infection with the Omicron variant and in vaccinated patients [23].

Furthermore, Schwartz et al. demonstrated the presence of the virus in placental tissue on the fetal side using in situ hybridization with RNAscore technology, thus establishing evidence of fetal exposure to SARS-CoV-2 and infection [24]. The consequences of maternal–fetal infection result from a virus-induced “cytokine storm”. There appears to be an excessive immune response responsible for placental insufficiency and transplacental transmission [25]. Indeed, SARS-CoV-2 causes massive inflammation by affecting the ciliated epithelium via angiotensin converting enzyme 2 (ACE2) receptors [26]. ACE2 is a component of the renin–angiotensin–aldosterone system which modulates blood pressure. This receptor is expressed by many tissues (cardiovascular, intestinal, adipose, pulmonary, renal, syncytiotrophoblastic, fetal, and neonatal airways). The virus binds to this receptor, which is abundant in syncytiotrophoblast cells from the end of the second trimester, and then enters them with the help of the transmembrane protease serine 2 (TMPRSS2), according to Hoffmann et al. [27]. Watkins et al. described typical lesions of placentitis that could be connected with SARS-CoV2 infection: association between perivillous fibrin deposits, chronic histiocytic intervillositis, trophoblast necrosis, and SARS-CoV-2 positivity of placentas [28]. Moreover, a case–cohort study in 2021 compared 27 placentas from pregnant women with asymptomatic or mildly symptomatic SARS-CoV-2 infection with 27 placentas from non-infected pregnancies. They found significantly greater maternal vascular malperfusion in the placenta of SARS-CoV-2-positive pregnancies (retroplacental hematomas, accelerated villous maturation, distal villous hyperplasia, fibrinoid necrosis, mural hypertrophy of membrane arteriole, vessel ectasia, and persistence of intramural endovascular trophoblast), without any significative difference in perinatal outcomes [29]. In our study, eight out of 15 placentas studied were infected by SARS-CoV-2 and six placentas presented the three histopathological lesions of placentitis. In other placentas, the association between trophoblast necrosis and fibrinoid deposits was missing in one of them and chronic histiocytic intervillositis was missed in the last one. SARS-CoV-2-induced histological lesions did not seem to affect placentas in accordance with the same risk factors as severe SARS-CoV-2 infections.

In a recent retrospective study, three fetal deaths and two extreme premature neonates were reported; the placentas showed massive perivillous fibrin deposition and large intervillous thrombi associated with strong SARS-CoV-2 expression in trophoblasts [30]. In our study, 88% neonates underwent iatrogenic prematurity, 70% were admitted to the NICU, 47% had fetal neonatal acidosis and one neonate died, which confirms consistent morbidity. Such poor neonatal outcomes could be related to the placental lesions due to SARS-CoV-2 infection. Placental histological abnormalities regardless of the severity of SARS-CoV-2 infection are responsible for hypoxia and NRHFR, which generate iatrogenic prematurity. Indeed, placental lesions are not specific to severe SARS-CoV-2 infection. In our study, as in previous studies, most women presented mild or moderate SARS-CoV-2 infection (88.7%), whereas they all underwent cesarean delivery [6,7,11]. Placental SARS-CoV-2 infection seems to be independent of the severity of maternal SARS-CoV-2 infection.

Up to now, our multicenter retrospective case description includes the largest number of described cases of NRFHR associated with biochemical disorders and placentitis in relation to maternal SARS-CoV-2 infection. However, our study presents several biases. First, it is a retrospective case series without comparison with a control group. The number of people recruited is moderate, but NRHFR outside labor and SARS-CoV-2 placentitis are rare events. This study is descriptive, and the inclusion criteria required a birth for NRFHR out of labor in the context of maternal infection with SARS-CoV-2. We observed that almost two thirds of the placentas showed characteristic histological signs of placentitis with COVID-19. We hypothesize that these lesions are at the origin of the abnormal placental exchange, and therefore of the NRFHR. Maternal infections with SARS-CoV-2 occurred within 10 days before birth, and thus the placental abnormalities observed. Because of the small number of patients (although the event is rare), it is not possible to clarify the relationship between the time of infection and the progression of placental pathology. Similarly, it may be possible that some situations of maternal SARS-CoV-2 infection may cause moderate and transient placental damage and therefore not cause NRFHR. However, the design of the study does not allow us to provide more information on this subject. In addition, there is a lack of data regarding SARS-CoV-2 variants, although it has been commonly accepted that the severity of SARS-CoV-2 infection may be related to the variant involved, especially the Alpha and Gamma variants which appear to be more morbid in the obstetrical context [31]. Finally, it is recognized that SARS-CoV-2 has significant neonatal morbidity due to spontaneous prematurity. Our case series allows us to line up one of the etiologies associated with an increase in induced prematurity.

## 5. Conclusions

Our cohort comprised 17 cases of SARS-CoV-2 placentitis, maternal biochemical abnormalities (coagulopathy, liver cytolysis), neonatal morbidity and mortality data. NRHFR and placental inflammatory lesions seem to be independent of the severity of maternal SARS-CoV-2 infection and maternal–fetal transmission. Indeed, the only risk factor associated with NRFHR was the lack of vaccination against SARS-CoV-2. We have not had any case of NRFHR since the end of the inclusion period, probably due to widespread vaccination and the appearance of variants such as the Omicron variant, which seems to be less pathogenic. Our results could help in the understanding of placental inflammatory lesions responsible for fetal distress.

## Figures and Tables

**Figure 1 viruses-15-01069-f001:**
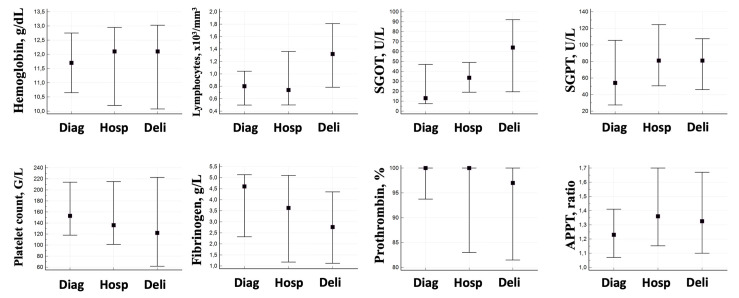
Maternal biochemical characteristics. Median (plain squares) and IQR (error bars) are provided. APTT: activated partial thromboplastin time. Diag: sampling at diagnostic. Hosp: sampling at hospitalization. Deli: sampling at delivery. SGOT: serum glutamic-oxaloacetic transaminase. SGPT: serum glutamic-pyruvic transaminase.

**Table 1 viruses-15-01069-t001:** Clinical characteristics of the population.

Clinical Characteristics (*n* = 17)	
Maternal age	36.0 (30.0–38.0)
Maternal BMI, kg/m^2^	25.0 (23.0–28.0)
Maternal BMI > 30 kg/m^2^	4/17 (23.5)
Geographic origin	
Europe	9/17 (52.9)
America	2/17 (11.8)
North Africa	4/17 (23.5)
South Africa	2/17 (11.8)
Previous pregnancies	
Nulliparous	4/17 (23.5)
Multiparous	13/17 (76.5)
Maternal comorbidities	
Smoking	2/17 (11.8)
Pregestational chronic hypertension	2/17 (11.8)
Pregestational diabetes mellitus	0
Pregestational liver disease ^1^	2/17 (11.8)
Asthma	3/17 (17.6)
Other ^2^	3/17 (17.6)
Complete anti-SARS-CoV-2 vaccination	0
Pregnancy characteristics	
Gestational diabetes	2/17 (11.8)
Moderate pre-eclampsia	1/17 (5.9)
Gravidic cholestasis	1/17 (5.9)

Median (IQR) or *n* (%) are provided. ^1^ Liver steatosis and chronical viral hepatitis B. ^2^ Thromboembolic event before pregnancy under anticoagulation, factor XI deficiency, MTHFR heterozygous mutation, sleeve-gastrectomy, hypothyroidism, previous surgery of aortic coarctation. BMI = body mass index.

**Table 2 viruses-15-01069-t002:** Characteristics of SARS-CoV-2 infection of the included women.

Characteristics of SARS-CoV-2 Infection (*n* = 17)	
Gestational age at diagnosis, weeks	32^6/7^ (30^5/7^–35^2/7^)
Symptoms	
Cough	8/17 (47)
Fever	12/17 (70.5)
Dyspnea	3/17 (17.6)
Headache	1/17 (5.8)
Nausea/vomiting	1/17 (5.8)
Myalgia	3/17 (17.6)
Anosmia/ageusia	2/17 (11.8)
WHO classification	
No symptoms	0
Mild	12/17 (70.5)
Moderate	3/17 (17.6)
Severe	2/17 (11.8)
Implicated viral strains	
Performed and reported (*n* = 7) Wild type (Wuhan-China) B.1.1.7 “Alpha” (UK strain) B.1.617.2 “Delta” (India)	1/7 (14.2) 5/7 (71.4) 1/7 (14.2)
Unknown	10/17 (58.8)
Obstetrical symptoms	
Uterine contractions	0
Decreased fetal movements	2/17 (11.8)
Premature rupture of membranes	0
Metrorrhagia	0
Hospitalization	
Interval between diagnosis and hospitalization (day, median IQR)	4 (2–6)
Gestational age at hospitalization	33^5/7^ (31^3/7^–35^6/7^)
Conventional hospitalization	15/17 (88.2)
Admission to ICU	2/17 (11.8)
Need for oxygen support	2/17 (11.8)
Orotracheal intubation	2/17 (11.8)

Median (IQR) or *n* (%) are provided. ICU: intensive care unit; SARS-CoV-2: Severe acute respiratory syndrome coronavirus 2; UK: United Kingdom; WHO: World Health Organization.

**Table 3 viruses-15-01069-t003:** Obstetrical and neonatal characteristics.

Obstetrical and Neonatal Characteristics	
Delivery (*n* = 17)	
Cesarean delivery	17 (100)
Gestational age at delivery, weeks	33^6/7^ (31^2/7^–35^4/7^)
Prematurity - 24–28 GW - 28–32 GW - 32–37 GW	1/17 (6) 5/17 (29) 9/17 (53)
Interval between hospitalization and delivery, days	5.0 (3.0–7.0)
Neonatal ouctomes	
Birthweight, g	2090 (1480–2200)
Birthweight, Z-score	−0.1 (−1.3–0.1)
Birthweight, Z-score (no, %) - Birthweight < 10th percentile - Birthweight < 5th percentile	5/17 (29) 2/17 (11.8)
<5 min Apgar score	6/15 (40)
Respiratory distress	11/17 (64.7)
NICU admission	12/17 (70.6)
Neonatal death	1/17 (5.9)
Umbilical cord pH (*n* = 15)	
Median, IQR	7.2 (7.1–7.3)
Acidosis (No., %) - pH between 7.00 and 7.20 - pH < 7.00	6/15 (40) 1/15 (7)
Neonatal SARS-CoV-2 RT-PCR by nasopharyngeal swab (*n* = 7)	
Positive	3/7 (42.8)
Negative	4/7 (57.1)
Placental analysis (*n* = 15)	
Placental SARS-CoV-2 identification (*n* = 8/15)	
Positive	8/15 (47)
Negative	0
Placentitis	
Fibrin deposition	10/15 (67)
Chronic intervillositis	10/15 (67)
Trophoblast necrosis	9/15 (60)

Median (IQR) or *n* (%) are provided. Percentiles of birthweight according the Z-score [17] and AUDIPOG [14]. NICU: neonatal intensive care unit.

## Data Availability

The data that support the findings of this study are available from the corresponding author upon reasonable request.

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
