# Peer review of "Impact of SARS-CoV-2 Infection on Unvaccinated Pregnant Women: Non-Reassuring Fetal Heart Rate Tracing Because of Placentitis"

_viruses, 2023, doi:10.3390/v15051069_

Round 1
Reviewer 1 Report
The authors of the manuscript "SARS-CoV-2 infection during pregnancy and non-reassuring fetal tracing outside of labor" present a nicely written paper which confirms previous studies. As the authors indicate, the multicentric study has limitations, mainly related to its retrospective nature. However, the angle presented is interesting, as they establish the connection between NRFHR and SARS-CoV-2 infection.
The novel angle is the correlation of placental histology and pregnancy outcome in mothers resulting in fetal delivery outside labor due to NRFHR. Regarding the Methodology, there is not much to add, as the case group was small (17 placentas) and it was a retrospective study.
The main question addressed by the research was to identify the clinical, biochemical, and histological determinants associated with the occurrence of non-reassuring fetal heart rate (NRFHR) outside labor in fetuses of SARS-CoV-2-infected mother. The confirmation of NRFHR outside labor is original, perhaps not so relevant to VIRUS as it would be for an Obstetric Journal. The specific gap addressed -NRFHR- is not so much relevant to VIRUS.
Apart from establishing a correlation between placental histology and pregnancy outcome in mothers with history of SARS-CoV-2 infection resulting in fetal delivery outside labor due to NRFHR, the study does not add any new information to what is already known.
Author Response
The authors of the manuscript "SARS-CoV-2 infection during pregnancy and non-reassuring fetal tracing outside of labor" present a nicely written paper which confirms previous studies. As the authors indicate, the multicentric study has limitations, mainly related to its retrospective nature. However, the angle presented is interesting, as they establish the connection between NRFHR and SARS-CoV-2 infection.
The novel angle is the correlation of placental histology and pregnancy outcome in mothers resulting in fetal delivery outside labor due to NRFHR. Regarding the Methodology, there is not much to add, as the case group was small (17 placentas) and it was a retrospective study.
The main question addressed by the research was to identify the clinical, biochemical, and histological determinants associated with the occurrence of non-reassuring fetal heart rate (NRFHR) outside labor in fetuses of SARS-CoV-2-infected mother. The confirmation of NRFHR outside labor is original, perhaps not so relevant to VIRUS as it would be for an Obstetric Journal. The specific gap addressed -NRFHR- is not so much relevant to VIRUS.
Thank you for your review and comments. We believe that the subject of our article could allow the medical community to become aware of the major impact of SARS-CoV-2 on maternal-fetal morbidity and mortality. A publication in Viruses would contribute to a better communication of our results and to raise awareness. We shall leave it to the Editor to see whether our publication, as we believe, has a place in Viruses and this special edition.
Apart from establishing a correlation between placental histology and pregnancy outcome in mothers with history of SARS-CoV-2 infection resulting in fetal delivery outside labor due to NRFHR, the study does not add any new information to what is already known.
There are very few described cases of non-reassuring fetal heart rate outside of labor in SARS-CoV-2. We have identified 11 cases in the bibliography. We therefore thought it would be interesting to share our collection of cases.
In addition, it is recognized that SARS-CoV-2 has significant neonatal morbidity due to spontaneous prematurity. Some studies suggest an increase in SARS-CoV-2 induced prematurity. Our case series allows us to line up one of the etiologies associated with this increase in induced prematurity and this seems important to us. We have clarified this issue in the discussion section.
Reviewer 2 Report
This article is an interesting report of SARS-CoV-2 infection during pregnancy and the prognosis for the mother and child together with placental pathology findings. Although the number of cases collected was 17, which limits statistical evaluation, the fact that the severity of maternal illness does not coincide with the severity of placental pathological changes is an important finding in the management of pregnant mothers and fetuses.
RESULTS
Although cases are presented as percentages only, it is better to present them as percentages (number of cases) because of the small number of cases.
Discussion
The following points should be addressed in the Discussion
1) The relationship between the timing of SARS-CoV-2 infection and the progression of placental pathology.
2) Are respiratory complications in newborns associated with prematurity or are they a direct disease of mother-to-child transmission of SARS-CoV-2?
3) Indicate any specific measures to be taken when a pregnant woman is infected with SARS-CoV-2
4) 4) How to accumulate a larger number of cases
Author Response
This article is an interesting report of SARS-CoV-2 infection during pregnancy and the prognosis for the mother and child together with placental pathology findings. Although the number of cases collected was 17, which limits statistical evaluation, the fact that the severity of maternal illness does not coincide with the severity of placental pathological changes is an important finding in the management of pregnant mothers and fetuses.
Thank you for your review and comments.
RESULTS
Although cases are presented as percentages only, it is better to present them as percentages (number of cases) because of the small number of cases.
Corrections have been made throughout the manuscript.
Discussion
The following points should be addressed in the Discussion
- The relationship between the timing of SARS-CoV-2 infection and the progression of placental pathology.
This study is descriptive and the inclusion criteria required a birth for NRFHR out of labour in the context of maternal infection with SARS-CoV-2. We observed that almost two thirds of the placentas showed characteristic histological signs of placentitis with COVID-19. We observed that almost two thirds of the placentas showed characteristic histological signs of COVID-19 placentitis. We hypothesise that these lesions are at the origin of the abnormal placental exchange, and therefore of the NRFHR. Maternal infections with SARS-CoV-2 occurred within 10 days before birth, and thus the placental abnormalities observed. Because of the small number of patients (although the event is rare), it is not possible to clarify the relationship between the time of infection and the progression of placental pathology. Similarly, it is possible that some situations of maternal SARS-CoV-2 infection may cause moderate and transient placental damage and therefore not cause NRFHR. But the design of the study does not allow us to provide more information on this subject.
- Are respiratory complications in newborns associated with prematurity or are they a direct disease of mother-to-child transmission of SARS-CoV-2?
Regarding the three newborns who were positive for SARS-CoV-2, all were hospitalized in the neonatal intensive care unit and presented with neonatal respiratory distress. The three fetuses had birth terms of 294/7 weeks, 33 weeks and 35 weeks. We cannot affirm that the respiratory distress they presented at birth was solely due to SARS-CoV-2 infection, given their prematurity term. In all cases, clinicians considered respiratory distress as a consequence of prematurity.
- Indicate any specific measures to be taken when a pregnant woman is infected with SARS-CoV-2
It is not possible for us to issue recommendations in the case of maternal infection by SARS-CoV-2 because 1/ we do not know the exact incidence of placental pathology associated with SARS-CoV-2 infections; 2/ the risk of iatrogeny must be taken into account in the event of the implementation of an active surveillance policy (induced prematurity, hospitalisation); 3/ the implementation of recommendations must be done within a rigorous scientific framework (evidence based medicine)
However, although these are not recommendations, we have provided some advice on monitoring measures. We specified these specific measures in our article at line 270 “In the case of maternal infection with SARS-CoV-2, the existence of biochemical abnormalities (transaminitis, coagulation abnormalities) could lead to consideration of the implementation of CTG monitoring to limit the risk of stillbirth”.
- How to accumulate a larger number of cases
As cited in our article at lines 100-104, collaboration was sought with the maternity hospitals in the CEGORIF (Cercle d’Étude des Gynécologues-Obstétriciens de la Région Ile-de-France), the APHP (Assistance Publique des Hôpitaux de Paris) and Burgmann University (Brussels) hospitals. The investigators were contacted by e-mail on three successive occasions over a period of one year. These events seem to be really rare, and only 11 cases had been published since 2020.
We also plan to open an international registry to identify cases and to better understand the pathophysiology of SARS-CoV-2 related placental damage.
Reviewer 3 Report
Claudet et al. compared the findings of foetal heart rate monitoring of COVID-19 infected mothers with those of placental pathology and blood biochemical findings in this manuscript. We deem this a significant topic, but it is insufficient for publication for the following reasons:
The number of cases is low (17 cases), but COVID-19 affects many pregnant women worldwide, and there have been numerous reports of fetoplacental dysfunction and stillbirths caused by placentitis. Noteworthy is the frequency of such a severe course among individuals affected.
2 Even when the mother is mildly ill, there have been reports of placentitis, which is not novel.
3 Fetal heart rate monitoring is performed on the vast majority of pregnant women in developed countries, with or without COVID-19 infection; therefore, it is necessary to clarify what characteristics of COVID-19 are observable. It is not specific enough to indicate findings of placental insufficiencies, such as placentitis or infarction. Exists data on the prevalence and incidence of delayed bradycardia and sinusoidal pattern?
4 The clinical picture is likely to vary depending on the virulence of the virus. The differences between the early δ and the latest o strains should be clarified.
5. The importance of vaccination and the lack of a correlation between maternal severity and foetal prognosis have already been established; therefore, the authors should present new clinical markers or suggestions.
Author Response
Claudet et al. compared the findings of foetal heart rate monitoring of COVID-19 infected mothers with those of placental pathology and blood biochemical findings in this manuscript. We deem this a significant topic, but it is insufficient for publication for the following reasons:
- The number of cases is low (17 cases), but COVID-19 affects many pregnant women worldwide, and there have been numerous reports of fetoplacental dysfunction and stillbirths caused by placentitis. Noteworthy is the frequency of such a severe course among individuals affected.
We believe that it is necessary to prevent and take measures before the unfavorable outcome of fetal death in utero. Only 11 cases of FHR abnormalities outside of labor in SARS-CoV-2 patients have been published to date, while millions of pregnant patients have been indeed infected with SARS-CoV-2 for the past 3 years. This discrepancy between the number of the infected population and the number of cases studied shows the difficulty of this census and the importance of making medical teams aware of this event.
As you pointed out, a link has been made between placental damage and stillbirth, but little data has been accumulated regarding “near misses” and the possibility of avoiding the risk of stillbirth. The purpose of this work was to highlight the intrinsic morbidity of SARS-CoV-2 infections in relation to placental damage, while not causing stillbirth. In this sense, it seemed important to be able to trace this series of cases and to describe correctly the clinical, biological and placental characteristics in these specific cases. We will leave it to the discretion of the Editor to disseminate, or not, this information which we considered important.
- Even when the mother is mildly ill, there have been reports of placentitis, which is not novel.
We focused our article on the joint association of moderate maternal disease, severity of FHR abnormalities requiring preterm birth and placental abnormalities. This point of view seemed original to us.
- Fetal heart rate monitoring is performed on the vast majority of pregnant women in developed countries, with or without COVID-19 infection; therefore, it is necessary to clarify what characteristics of COVID-19 are observable. It is not specific enough to indicate findings of placental insufficiencies, such as placentitis or infarction. Exists data on the prevalence and incidence of delayed bradycardia and sinusoidal pattern?
Fetal heart rate abnormalities in SARS-CoV-2 infection are non-specific and differ among the cases we studied. However, all FHR abnormalities were severe enough to lead to premature extraction by cesarean section (catergory II ou III according to ACOG classification). We believe that the abnormalities of the fetal heart rate observed in the event of SARS-CoV-2 infection are nevertheless specific to our cases, knowing that no obstetric pathology was involved. Moreover, the association between infection with SARS-CoV2 and stillbirths is now known. In our study, we also found 3 cases of maternal-fetal transmission where the mother, the fetus and the placenta were infected with SARS-CoV-2, without any other obstetric pathology explaining the induced prematurity. In addition, almost no obstetric pathology is associated with the occurrence of specific fetal heart rate abnormalities (with the possible exception of the sinusoidal rhythm of severe and prolonged fetal anemia).
- The clinical picture is likely to vary depending on the virulence of the virus. The differences between the early δ and the latest o strains should be clarified.
Unfortunately, it is not possible to elaborate on this because, as explicitly mentioned in the limitations of this work, variant sequencing was not performed for the vast majority of patients. Moreover, even if we had this information, we generated a selection bias that we assume by the methodology of the article. A study should have been carried out with a control group composed of patients without fetal heart rhythm abnormalities. We wanted to focus on purely describing a cohort of well-characterised clinical situations without a control group.
- The importance of vaccination and the lack of a correlation between maternal severity and foetal prognosis have already been established; therefore, the authors should present new clinical markers or suggestions.
The methodology of this study does not allow us to evaluate new clinical markers as it is a retrospective study on routine care data. On the other hand, this study seems to provide a new insight into the pathophysiology of maternal SARS-CoV-2 infections on placental damage and its potential consequences on neonatal morbidity (induced prematurity / neonatal acidosis.As mentioned above, we plan to set up an international prospective registry to answer the remaining questions raised by this work.
Round 2
Reviewer 1 Report
The study has limitations related to the small number of retrospective nature. In the added paragraph under Discussion, there is a repeated sentence (bottom of page 9)